# Predicting Tumor Perineural Invasion Status in High-Grade Prostate Cancer Based on a Clinical–Radiomics Model Incorporating T2-Weighted and Diffusion-Weighted Magnetic Resonance Images

**DOI:** 10.3390/cancers15010086

**Published:** 2022-12-23

**Authors:** Wei Zhang, Weiting Zhang, Xiang Li, Xiaoming Cao, Guoqiang Yang, Hui Zhang

**Affiliations:** 1Department of Urology, First Hospital of Shanxi Medical University, Taiyuan 030001, China; 2College of Medical Imaging, Shanxi Medical University, Taiyuan 030001, China; 3Department of Radiology, First Hospital of Shanxi Medical University, Taiyuan 030001, China; 4Intelligent Imaging Big Data and Functional Nano-Imaging Engineering Research Center of Shanxi Province, First Hospital of Shanxi Medical University, Taiyuan 030001, China

**Keywords:** prostate cancer, PNI, bi-parametric MRI, radiomics, nomogram

## Abstract

**Simple Summary:**

Perineural invasion (PNI) is present in 17–75% of prostate cancer patients and is an important mechanism for cancer progression, leading to poor prognoses. An optimized preoperative technique is needed to detect PNI in prostate cancer patients and administer the best treatment. The aim of our retrospective study was to develop a model based on high-throughput radiomic features of bi-parametric MRI combined with clinical factors that can predict PNI status in high-grade prostate cancers. In total, 183 high-grade PCa patients were included in this retrospective study, and the radiomics model based on 13 selected features of bi-parametric MRI showed better discrimination than did the conventional model in the test cohort (area under the curve (AUC): 0.908). Discrimination efficiency improved when the radiomics and clinical models were combined (AUC: 0.947). This improved model may help predict PNI in prostate cancer patients and allow more personalized clinical decision-making.

**Abstract:**

Purpose: To explore the role of bi-parametric MRI radiomics features in identifying PNI in high-grade PCa and to further develop a combined nomogram with clinical information. Methods: 183 high-grade PCa patients were included in this retrospective study. Tumor regions of interest (ROIs) were manually delineated on T2WI and DWI images. Radiomics features were extracted from lesion area segmented images obtained. Univariate logistic regression analysis and the least absolute shrinkage and selection operator (LASSO) method were used for feature selection. A clinical model, a radiomics model, and a combined model were developed to predict PNI positive. Predictive performance was estimated using receiver operating characteristic (ROC) curves, calibration curves, and decision curves. Results: The differential diagnostic efficiency of the clinical model had no statistical difference compared with the radiomics model (area under the curve (AUC) values were 0.766 and 0.823 in the train and test group, respectively). The radiomics model showed better discrimination in both the train cohort and test cohort (train AUC: 0.879 and test AUC: 0.908) than each subcategory image (T2WI train AUC: 0.813 and test AUC: 0.827; DWI train AUC: 0.749 and test AUC: 0.734). The discrimination efficiency improved when combining the radiomics and clinical models (train AUC: 0.906 and test AUC: 0.947). Conclusion: The model including radiomics signatures and clinical factors can accurately predict PNI positive in high-grade PCa patients.

## 1. Introduction

Prostate cancer (PCa) is the most frequent malignant tumor in 105 countries worldwide and the first leading cause of cancer-related death in 46 countries among males [1]. Often, there are significant differences in the prognosis of patients with the same stratification who adopt the same treatment plan [2]. In addition, many localized PCa cases, especially high-grade cases, are not truly localized tumors when they are diagnosed. The reasons for this situation are that cancer cells have already spread beyond the scope of surgery or radiotherapy, and these patients are prone to developing biochemical recurrence [3]. It is widely accepted that prostate-specific antigen (PSA), Gleason score (GS), and T stage are the main variables for evaluating the prognosis of localized PCa. Among the factors causing tumor spread, perineural invasion (PNI), which is invasion along or around nerves within the perineural space, also plays an important role in cancer [4]. PNI can be evaluated in a biopsy specimen or radical prostatectomy specimen, and it is present in 17–75% of prostate cancer patients [5]. The College of American Pathologists published a consensus statement on prognostic factors for PCa in which PNI was identified as a potential prognostic factor (category III) that needed additional study [6]. Therefore, identifying the PNI status of high-grade PCa is an urgent problem to be solved.

At present, magnetic resonance imaging (MRI) is widely used for diagnosing PCa and can help detect several prognostic factors; it has been used to increase T staging accuracy and predict positive surgical margins (PSMs) by detecting and localizing extracapsular extension (ECE) [7,8]. Radiomics, as an extension concept of texture analysis, can convert medical images into high-dimensional mineable and quantitative features by using high-throughput extraction algorithms of these characterizations. In recent years, qualitative analysis of prostate MRI images by means of radiomics plays a crucial role at the pretreatment staging step and is increasingly applied to determine invasion and prognosis for prostate cancer [9,10]. PNI is a pathological feature that can only be detected after an invasive biopsy or prostatectomy. This form of metastasis can affect peri-prostatic neurovascular fibers, the lumbosacral plexus, and the sciatic nerve, and MRI can visualize involvement of these nerve fibers as direct evidence of cancer cell spreading [11,12]. In the age of high-resolution imaging, developing a method based on radiomics to accurately assess the PNI status of PCa is urgently needed.

In this study, we evaluated the relationship between MRI radiomics signature, as well as other clinical and pathological factors, and PNI in high-grade PCa. We hypothesized that the MRI radiomics signature may provide effective information and established a model for preoperatively predicting the probability of PNI in high-grade PCa patients.

## 2. Materials and Methods

### 2.1. Patients

This retrospective study received Institutional Review Board approval of the First Hospital of Shanxi Medical University, ethic code: (K131). We retrospectively selected PCa patients with clinical and imaging data from January 2016 to May 2021 who underwent prostate MR examination before systematic prostate biopsy or radical prostatectomy (RP). Clinical data, including age, PSA level, prostate volume, prostate-specific antigen density (PSAD), GS, grading groups (GGs), and tumor location in the prostate, were collected from patient medical records. The study inclusion criteria were as follows: (a) high-grade PCa patients who underwent prostate MRI examination; and (b) tumor perineural invasion status obtained on histopathology by biopsy or RP. The following exclusion criteria were applied: (a) PCa patients who received other treatments before MRI examination, such as androgen suppression therapy or any previous transurethral surgery; (b) poor image quality due to artifacts; (c) incomplete MR sequence; and (d) incomplete clinical data collection; (e) the lesions were too small for segmentation and analysis (maximum diameter <3 mm). A total of 208 high-grade prostate cancer patients’ data were collected. According to the exclusion criteria, 25 patients were excluded. Ultimately, 183 high-grade PCa patients were enrolled in the study. The patients were randomly divided into training and test groups at a ratio of 7 to 3 (training group: 128 patients, test group: 55 patients).

### 2.2. MR Image Data

The prostate MRI examination was performed according to PI-RADS v2.1 protocol and the process was as follows. We utilized a 3.0-T scanner (GE Signa HDxt) with an 8-channel array coil to acquire the images of multiplanar T2-weighted imaging (T2WI) and diffusion-weighted imaging (DWI), which were obtained with a turbo spin‒echo sequence and the following parameters: repetition time/echo time (TR/TE): 3360/68.16 ms; field of view (FOV): 220 × 220 mm; matrix: 320 × 256; slice thickness: 5 mm; and spacing between slices: 5.5 mm. A single-shot echo-planar sequence with four b-values was also acquired: 0 and 1500 s/mm (TR/TE: 5250/78.6 ms; FOV: 100 × 100 mm; matrix: 128 × 160; and slice thickness: 5 mm).

### 2.3. Histopathologic Analysis

All patients underwent transrectal ultrasound-guided 12-core systematic prostate biopsy or RP after prostate MRI examination. The specimen pathological diagnosis was made by two pathologists with more than three years of experience in diagnosis of prostate diseases. The GS was updated according to the 2014 International Society of Urological Pathology criteria. PNI was diagnosed when PCa infiltration was identified in any layer of the nerve sheath or tumor invasion involved at least one-third of the nerve circumference. Pathologic information was collected, and, according to the outcomes, all patients were divided into two groups: one group had positive prostate cancer cell PNI and the other group had negative prostate cancer cell PNI (Figure 1).

### 2.4. Tumor Segmentation

All MR images were manually delineated by two independent readers with more than 5 years’ experience in reading prostate MR images. ITK-SNAP software was used to process T2WI and high-b-value (b = 1500) DWI images. Tumors were targeted as the regions of interest (ROIs), defined as hypointense signal areas compared with the normal prostate area on T2WI and a higher signal intensity than that of the normal prostate area on DWI. For consistency between ROIs in both T2WI and DWI images, all depicted ROIs were strictly delineated with the same criteria and visually validated by the same expert. The ROIs were manually delineated layer-by-layer along the lesion boundary, obtaining three-dimensional data (Figure 1).

### 2.5. Extraction of Radiomic Features

Software of FAE (FAE version is 0.5.2 and PyRadiomics version is 3.0.1. The software was soured from East China Normal University, Shanghai, China. https://github.com/ salan668/FAE accessed on 16 December 2022), which was developed based on the PyRadiomics package (https://github.com/Radiomics/pyradiomics, accessed on 2 June 2022), was used to extract features from the T2WI ROIs and DWI ROIs. The parameters of feature extraction were: first order statistics, shape-based, GLCM, GLRLM, GLSZM, GLDM, NGTDM. A total of 1702 features were extracted from the MRI data and 851 features each from T2WI and DWI, including 14 shape features, 18 first-order features, 24 gray level co-occurrence matrix (GLCM) features, 16 gray level run length matrix (GLRLM) features, 16 gray level size zone matrix (GLSZM) features, 5 neighboring gray tone difference matrix (NGTDM) features, and 14 gray level dependence matrix (GLDM) features and 744 wavelet features [13].

### 2.6. Feature Selection and Model Building

The process of feature selection was based on training set. Thirty patients were randomly selected for a double-blinded comparison of manual segmentations by two radiologists. Inter- and intraclass correlation coefficients (ICCs) between groups and within groups were calculated to select features with high stability and reproducibility, and ICCs greater than or equal to 0.75 were considered to have good agreement. To remove the imbalance of the training dataset, we used the synthetic minority oversampling technique (SMOTE) to balance the positive/negative samples. Before feature selection, we subtracted by the mean value and divided by the standard deviation to normalize the feature matrix for each feature vector. Next, the feature selection process was divided into two steps. In the first step, the features with statistical significance for identifying PNI positivity were selected by univariate logistic regression analysis. In addition, the first stage of dimensionality reduction of the data was achieved to ensure that each feature had a significant effect on the outcome. In the second step, least absolute shrinkage and selection operator (LASSO) regression analysis was used for further data dimensionality reduction, and the best features were determined for establishment of the radiomics model. The hyperparameter lambda value and the number of selected features were determined by tenfold cross-validation. After the radiomics model was established, each feature was multiplied by its corresponding coefficient, and an intercept value was added to calculate the radiomics score (Rad-score) for each patient, which was establishment of the radiomics signature (Appendix A).

For clinical features, we used the univariate analysis method, and the features with statistical significance for the results were selected to construct a clinical model. Finally, the combined model of clinical and radiomics features was established by multiple logistic regression analysis method.

### 2.7. Model Evaluation

After the models were built, their performance was evaluated using receiver operating characteristic (ROC) curve analysis. The area under the ROC curve (AUC) was calculated for quantification of the performance. The accuracy, sensitivity, and specificity were also calculated at a cutoff value that maximized the value of the Youden index. A radiomic nomogram combining the Rad-score derived from T2WI and DWI scans and clinical factors was developed for predicting PNI. The calibration curves measured the consistency between the predicted probability of PNI and the actual probability of PNI. Decision curve analysis was applied to measure the clinical utility of the nomogram.

### 2.8. Statistical Analysis

Demographic data were compared by chi-squared test, Mann‒Whitney test, or *t*-test. Continuous variables are expressed as mean ± standard deviation, and categorical variables are expressed as median (25 quantile, 75 quantile). A value of *p* < 0.05 was considered statistically significant. Statistical analyses were performed using SPSS v22.0 (IBM SPSS Statistics, IBM Corp., Armonk, NY, USA) and R software (R is a language and environment for statistical computing and graphics. It is a GNU project which is similar to the S language and environment which was developed at Bell Laboratories (formerly AT&T, now Lucent Technologies) by John Chambers and colleagues, version 4.1.2; http://www.Rproject.org, accessed on 17 December 2022).

## 3. Results

### 3.1. Patient Characteristics

PNI was diagnosed histologically based on RP or biopsy specimen tissues. In total, 183 patients were then divided into the PNI positive [PNI (+)] group and the PNI negative [PNI (−)] group. The PNI (+) group contained 54 patients (29.51%), while the PNI (−) group contained 129 patients (70.49%). In the PNI positive group, 42 were detected on RP and 12 on biopsy. Twenty-seven of the forty-two cases were confirmed PNI positive both on preoperative biopsy and RP; eight of the forty-two cases had no PNI positive results on biopsy, but the RP outcomes were determinative; seven of the forty-two cases obtained a biopsy at another center, and we only had PNI positive results after RP in our center. Twelve PNI positive cases confirmed by biopsy did not undergo RP after biopsy in our center. The concordance rate of PNI positive results between biopsy and RP was 64.29%. In the PNI negative group, 98 cases were diagnosed as PNI negative both on preoperative biopsy and RP; 31 cases obtained a biopsy at another center; we only had their PNI negative outcomes of RP in our center. The concordance rate was 75.97%. The average ages were 69.7 ± 8.2 years and 72.0 ± 9.0 years in the two respective groups. The PSA levels were 15.9 ng/mL and 17.4 ng/mL in the two respective groups. In the PNI (+) group, the GS proportions were distributed as follows: 22.2% of patients (12/54) had a score of 8, 42.6% (23/54) had a score of 9, and 11.1% (6/54) had a score of 10. In the PNI (−) group, the GS proportions were distributed as follows: 41.1% of patients (53/129) had a score of 8, 39.5% (51/129) had a score of 9, and 19.4% (25/129) had a score of 10. The radiological and other clinical characteristics of the two groups are summarized in Table 1. There were no significant differences between these two groups in terms of age, PSA level, PSAD, or tumor location. However, there were significant differences in prostate volume, GS, and GG (*p* < 0.05). There were no significant differences between the training and test cohorts in terms of all clinical characteristics, which are summarized in Table 2 (*p* > 0.05).

### 3.2. Feature Selection and Comparison of Models

Further, 1193 stable features with ICCs ≥ 0.75 were retained (611 features from T2WI, and 582 features from DWI). The T2WI sequence selected 10 features when the λ_1se_ was equal to 0.06478 and obtained the highest AUC on the testing dataset. The AUC and accuracy of the model were 0.827 (95% CI 0.707–0.947) and 0.818, respectively. The DWI sequence selected four features when the λ_1se_ was equal to 0.11225 and obtained the highest AUC on the testing dataset. The AUC and accuracy of the model were 0.734 (95% CI 0.593–0.975) and 0.746, respectively. The T2WI + DWI sequence selected 13 features when the λ_1se_ was equal to 0.06787 and obtained the highest AUC on the validation dataset. The AUC and accuracy of the model were 0.908 (95% CI 0.821–0.996) and 0.855, respectively. Thirteen features were found to have high stability for prediction of PNI and were chosen to construct the final model. The details of feature selection and comparison of models were shown in Figure 2 and Figure 3 and Table 3 and Table 4.

The clinical model based on features including FH, RL, prostate volume, and GS obtained the highest AUC on the test dataset. The AUC and accuracy of the model were 0.823 (95% CI 0.712–0.933) and 0.673, respectively, on the testing dataset (Figure 2 and Figure 3 and Table 4).

### 3.3. Development of the Clinical–Radiomics Predictive Model

After the independently associated risk factors of FH, RL, volume, and GS were selected, we combined them with the Rad-score of the 13 features to form a PNI predictive nomogram. This nomogram had better performance in predicting PNI: the AUCs were 0.906 (95% CI 0.866–0.947) in the training group and 0.947 (95% CI 0.884–1) in the test group (Figure 4 and Table 4).

### 3.4. Validation of the Clinical–Radiomics Predictive Nomogram

The calibration charts showed that the actual probability of PNI occurrence was consistent with the predicted probability, and the Hosmer‒Leme show test yielded P values of 0.907 and 0.689 in the training and test cohorts, respectively. As shown in Figure 5, decision curve analysis indicated that the PNI predictive nomogram model was the best method across the full range of reasonable threshold probabilities. In the training group, the net reclassification index (NRI) was 1.1252 (0.8659–1.3644, *p* < 0.01) comparing the clinical model and combined model, while the NRI was 0.886 (0.6271–1.449, *p* < 0.01) comparing the radiomic model and combined model. In the test group, the NRI was 1.2312 (0.7796–1.6829, *p* < 0.01) comparing the clinical model and combined model, while the NRI was 1.0691 (0.5958–1.5424, *p* < 0.01) comparing the radiomic model and combined model (Figure 6).

## 4. Discussion

PNI is a histological phenomenon in which cancer cells surround and invade nerves in the tumor microenvironment and play a role in development and regeneration of cancer cells. Nerves and cancer cells communicate bidirectionally to each other, providing a mechanism that could induce cancer invasion and spread. Studies have shown that the sympathetic nervous system in cancer can regulate pathological gene expression, leading to DNA damage repair inhibition and oncogene activation to increase cancer cell metastasis and tumorigenesis [14,15]. On the other hand, cancer cells can secrete neurotrophic growth factors or chemokines, such as CCL2 and CXCL12, to promote development of neural progenitors, causing nerve growth [16,17]. PNI in cancer is associated with poor prognosis, likely because neoplastic cells hidden in the perineural space cannot be removed during tumor resection and cause recurrence.

In 1999, the College of American Pathologists published a consensus statement on prognostic factors for PCa in which PNI was classified as category III for risk of recurrence and needed additional study [6]. In multivariate analysis, PNI on biopsy showed significance for recurrence. The presence of PNI on target-biopsy associated with worse histopathologic features on RP and poorer outcomes might thus be useful for risk stratification [18]. As primary treatment decisions are often based on biopsy results, the additional PNI information may be relevant for optimal patient care [19]. PNI found on prostate biopsies has been shown to be an independent predictor of high-grade disease associated with a higher mean PSA, adverse pathologic features of higher GS, and extra-prostatic extension [20,21]. In our study, 54 PNI (+) patients among 183 high-grade PCa patients had higher GG and GS than PNI (−) patients, and the outcome was consistent with these studies. PCa patients with PNI positivity showed an increased risk of biochemical recurrence after prostatectomy or radiotherapy and worse survival outcomes, which have important implications for treatment decision-making and management of PCa [22,23,24].

The slowly progressive nature of nerve involvement can often make PNI difficult to diagnose, and PNI is always detected based on the pathological results of the biopsy and prostatectomy specimens of PCa patients. As not all PCa cases are diagnosed at the initial biopsy, PNI as an independent prognostic factor remains difficult to quantitatively measure in pathological samples because of its heterogenous presentations and the multifocal nature of RP specimens [25]. Recent research has shown that the distribution of nerves within the tumor-infiltrating microenvironment is not homogeneous. The neural density was significantly higher in the cancer periphery close to cancer infiltration than in the cancer core area, which suggests that nerves may drive tumor progression and invasion [26]. Many factors may influence the true pathological positive rate of PNI, such as the needle core number of biopsy and the processing method of RP specimen tissues [27]. Thus, the prognostic value of PNI evaluation in pathological analysis should be further assessed and a better method should be developed to provide a detailed spatial representation of heterogeneity.

MRI is a noninvasive diagnostic tool that can acquire entire anatomical images of the prostate for cancer staging, such as extra-prostatic extension. This is important for urologists to determine a treatment plan before surgery, such as preservation of the neurovascular bundle (NVB) [28]. In the era of high-resolution imaging, extra-prostatic extension on MR images already has a better ability to predict locally advanced-stage PCa than PNI positivity on biopsy [29]. Whether PNI, as a predominant mechanism and a predictor of PCa progression to an advanced stage, can be directly assessed on imaging measures needs further study to develop a visualization method. Jonathan J. Stone retrospectively reviewed the data of 3733 PCa patients from a medical database who had undergone both MRI and PET before surgery to identify direct radiological evidence of PNI. Fifteen patients who had perineural spread found on MRI presented enlargement of the spinal nerves, lumbosacral plexus, sciatic nerve on T1-weighted sequences, hyperintensity on T2-weighted sequences, and/or abnormal nerve enhancement after gadolinium administration [30]. Salvatore Siracusano evaluated a new MRI modality called diffusion tensor imaging (DTI), which can provide sharp depiction of peripheral nervous fibers to detect changes in peri-prostatic neuro-vasculature (PNF) before and after RP. DTI was able to detect quantitative changes in the number, length, and fractional anisotropy values of the PNF, and they observed that the fiber number in MRI images can serve as a recovery indicator of erectile dysfunction in nerve-sparing prostatectomy [31]. However, PNI is a microscopic-level finding in PCa. Huijuan You combined MRI and magnetic particle imaging involving superparamagnetic iron oxide nanoparticles to precisely distinguish high and low nerve densities of the PCa tissue microenvironment in a mouse model. Their method could visualize the nerve density, and they observed a positive correlation with the aggressiveness of PCa cancer cells, which can be a novel strategy for discovering biomarkers for neural tissue and tumor aggressiveness in PCa [32].

Although MR plays an important role in detecting and accurately evaluating PCa, image outcome reporting depends on the subjective judgment of radiologists, which causes high inter-reader variability. Recently, the quantitative analysis method based on machine learning techniques called radiomics was shown to automatically obtain high-throughput imaging features to overcome the above limitations and assess tumor biology characteristics. Several studies have reported use of MR-based radiomics to detect clinically significant PCa and assess aggressiveness and tumor staging [33]. Shuai Ma developed and validated a radiomics model that contains 17 stable radiomics features extracted from 1619 features based on T2WI to predict ECE in PCa. The AUC was 0.883 in the validation cohort, and the model was more sensitive than the radiologists’ interpretations, especially for apical tumors, which would influence a nerve-sparing surgical plan [34]. PNI is a predominant mechanism of ECE in PCa; to the best of our knowledge, there is no radiomics model based on MRI for preoperatively predicting this histopathological phenomenon.

In our study, we constructed a model derived from clinical and imaging data, including radiomic features from T2WI and DWI, based on computer-aided analysis to evaluate the PNI status in high-grade PCa. Our best radiomics model contained three GLDM features, one GLRLM feature, two NGTDM features, three GLSZM features, two GLCM features, one first-order feature, and one shape feature from T2WI and DWI images, which have the best predictive ability for PNI status in high-grade PCa. Our results demonstrated that the NGTDM feature had the greatest weight of the features in the T2WI model, while, in the DWI model, it was the GLCM feature, which is associated with tumor invasion and is a predictor of PCa aggressiveness, consistent with recently published findings concerning risk stratification for Pca. This finding suggests that invading nerves in the tumor microenvironment may affect the homogeneous texture features and that these radiomics features associated with PNI positivity may provide some additional information related to Pca aggressiveness, as previous studies reported [35,36]. The feature with the greatest weight in the T2WI + DWI model was the higher-order feature GLDM; this feature describes the gray level intensity within the ROI between the PNI positive and PNI negative groups and is used to highlight local heterogeneity information. This texture feature was rarely mentioned in previous radiomics studies for Pca, but, for other tumors, such as rectal cancer and cervical cancer, GLDM was thought to be associated with locally advanced tumors and poor prognosis in recent studies [37,38]. Similar to those in nontumor tissues, the GLDM metrics were found to be significantly different among peritumoral fat between high-grade and low-grade clear cell renal carcinoma and urothelial carcinoma [39,40]. Therefore, whether radiomics feature GLDM could be a biomarker for predicting the heterogeneity of interstitial composition in urologic cancers requires more research. Similar to the study of B. De Santi, which showed that a difference in voxel intensity distribution could distinguish cancerous and normal prostatic tissues [41], our model led to the conclusion that differences in heterogeneity between PNI positive and PNI negative samples can be detected and, therefore, can help depict the tissue microstructure as PNI positive or PNI negative before surgery.

Our clinical–radiomics prediction model, which integrates clinical characteristics and the Rad-score derived from MRI, had good sensitivity (0.944) and good specificity (0.865) in the test cohort, indicating that it is superior to all the above-mentioned models for predicting PNI status. Comparing the AUC values in the independent test cohort, our clinical–radiomics prediction model (AUC 0.947; 95% CI 0.884–1) performed better than the radiomics model alone (AUC 0.908; 95% CI 0.821–0.996) and the clinical model alone (AUC 0.823; 95% CI 0.712–0.933). Decision curve analysis showed that the clinical–radiomics model had a better ability to predict PNI than the other two models at any given threshold probability. This finding confirms that assessment of PNI with clinical or radiomic information alone will not be comprehensive.

Several limitations should be noted when considering this study. First, we included GGs of high-grade patients only; those with GS ≤ 7 patterns were excluded, especially patients with GS 4 + 3 who have a much worse prognosis, and their PNI status was not assessed. Second, some GS values were based on biopsy rather than on RP in our study, possibly causing sampling error. Third, there was a lack of spatial co-registration of the histopathology slides and MR images, which may cause a mismatch in delineating the ROIs directly on the T2WI and DWI images. Fourth, FAE software can be used conveniently for binary classification, but it has not yet provided an integrated UI for multilabel classification and regression problems. Fifth, this study was a single-institutional retrospective study design without external validation.

## 5. Conclusions

In our study, the results showed that MRI-derived radiomic features can be independent predictors of PNI in high-grade PCa. The combination of radiomic features extracted from T2WI and DWI maps produced higher diagnostic power to predict PNI than a single pattern. Additionally, our clinical–radiomics model was superior to a single radiomics model and a clinical model, suggesting that, combined, the radiomic features and clinical pathology information may have considerable value in predicting PNI in high-grade PCa, which can aid clinicians in choosing appropriate treatment options and estimating prognoses for such patients.

## Figures and Tables

**Figure 1 cancers-15-00086-f001:**
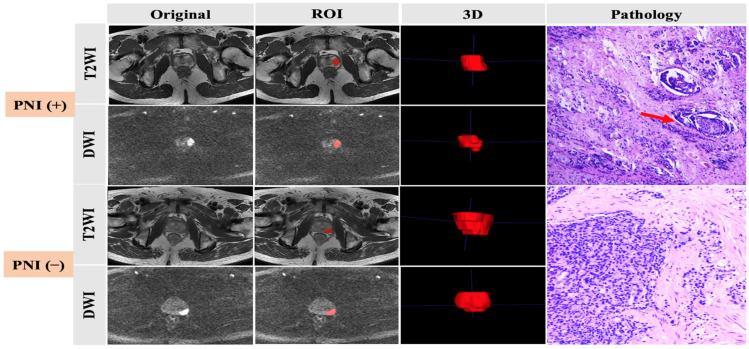
Preoperative MRI images, ROI delineation, and pathological comparison of prostate cancer with and without PNI, as indicated by the arrow.

**Figure 2 cancers-15-00086-f002:**
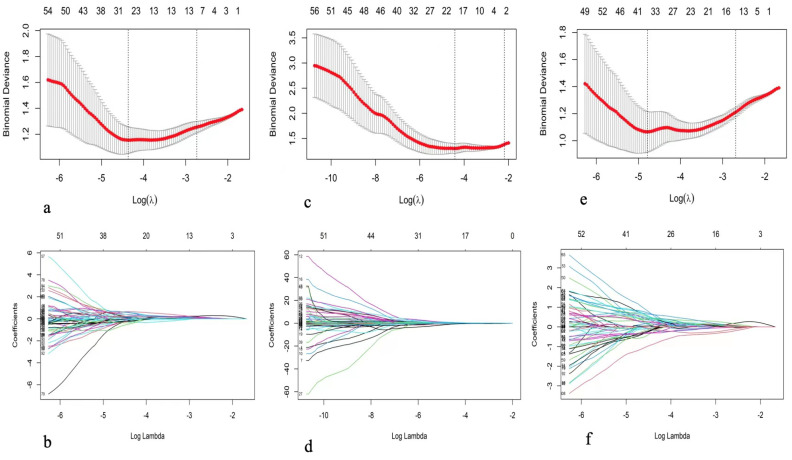
The lasso plots for radiomics feature selection: (**a**,**b**) for T2WI, 10 features were selected when the λ_1se_ = 0.06478, (**c**,**d**) for DWI, 4 features were selected when the λ_1se_ = 0.11225, and (**e**,**f**) for T2WI + DWI sequences, 13 features were selected when the λ_1se_ = 0.06787.

**Figure 3 cancers-15-00086-f003:**
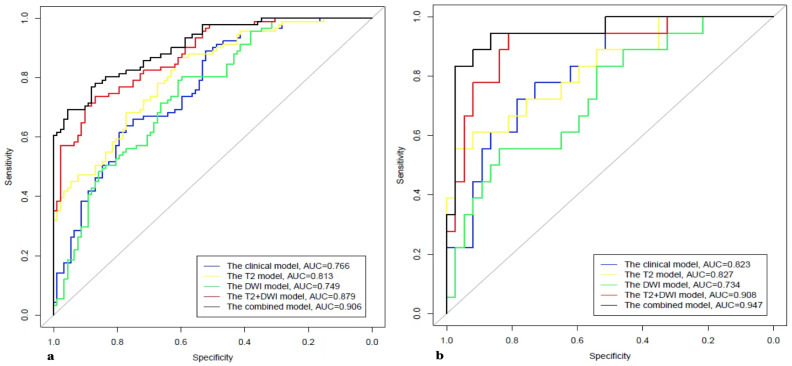
The AUCs of different models in the training (**a**) and test (**b**), respectively.

**Figure 4 cancers-15-00086-f004:**
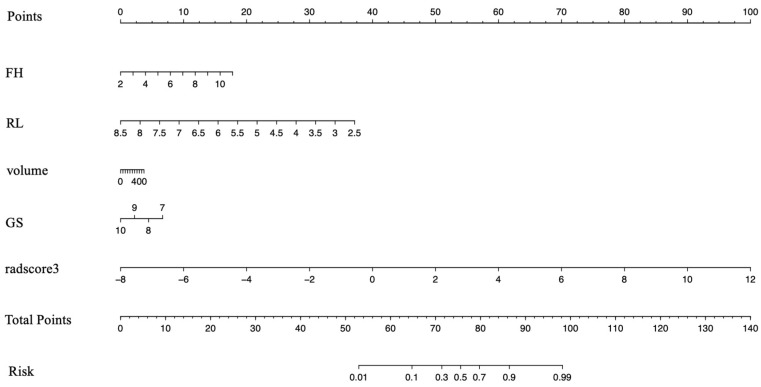
Nomogram developed for prediction of PNI. Radiomic nomogram combining the Rad-score derived from T2WI and DWI scans and clinical–radiological factors for predicting PNI. PNI: perineural invasion.

**Figure 5 cancers-15-00086-f005:**
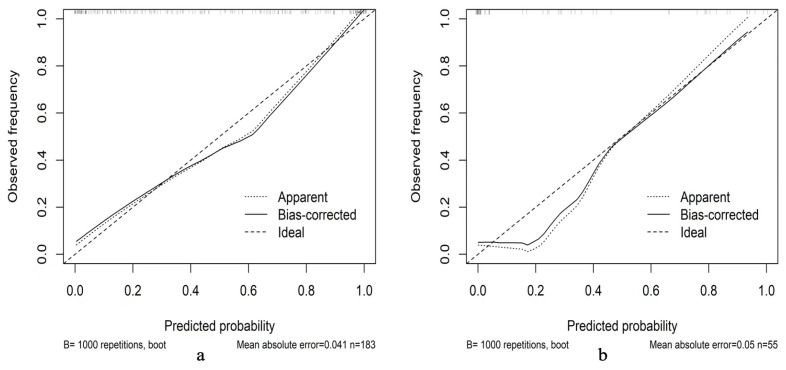
Calibration curve of the nomogram in the training (**a**) and test (**b**) groups.

**Figure 6 cancers-15-00086-f006:**
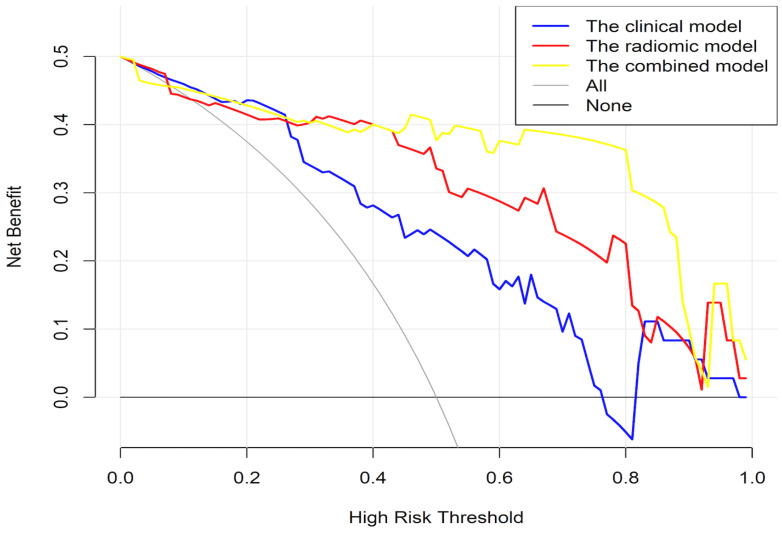
Decision curve analysis.

**Table 1 cancers-15-00086-t001:** Patient clinic radiological characteristics between groups of PNI (+) and PNI (−).

Characteristics	PNI (+) (N = 54)	PNI (−) (N = 129)	*p* Value
Age (years)	69.7 ± 8.2	72.0 ± 9.0	0.121
PSA level (ng/mL)	15.9 (10–23)	17.4 (11.4–25.7)	0.406
Prostate volume (mL)	43.7 (31.3–59.7)	53.7 (38.1–87.7)	0.006
Foot–head (FH) (cm)	4.4 (3.6–5.1)	4.7 (3.9–5.8)	0.02
Right–left (RL) (cm)	4.7 (4–5)	5.1 (4.5–5.9)	<0.001
Anterior–posterior (AP) (cm)	4.1 (3.6–4.9)	4.3 (3.7–5.2)	0.247
PSAD (ng/mL/cm^3^)	0.4 (0.2–0.5)	0.3 (0.2–0.5)	0.176
Gleason Score (GS)	9.13 (9–10)	8.78 (8–9)	0.005
Grading Groups (GG)			<0.001
Grade 1	0.0% (0/54)	0.0% (0/129)	
Grade 2	0.0% (0/54)	0.0% (0/129)	
Grade 3	0.0% (0/54)	0.0% (0/129)	
Grade 4	22.2% (12/54)	41.1% (53/129)	
Grade 5	77.8% (42/54)	58.9% (76/129)	
Location			0.196
Central zone	1.9% (1/54)	2.3% (3/129)	
Transition zone	13.0% (7/54)	7.0% (9/129)	
Peripheral zone	25.9% (14/54)	17.1% (22/129)	
Multiple zone	59.3% (32/54)	73.6% (95/129)	
Rad-score	1.52 ± 2.649	−1.815 ± 2.065	<0.001

**Table 2 cancers-15-00086-t002:** Patient clinic radiological characteristics between training and test cohort.

Characteristics	Training (N = 128)	Test (N = 55)	*p* Value
Age (years)	72.0 ± 8.6	69.8 ± 9.1	0.117
PSA level (ng/mL)	42.4 (14.3–138.6)	49.8 (13.9–169)	0.716
Prostate volume (mL)	48.6 (35.2–77.4)	52.9 (36.6–71.0)	0.797
Foot–head (FH) (cm)	4.7 (3.8–5.7)	4.6 (3.8–5.3)	0.484
Right–left (RL) (cm)	4.9 (4.4–5.5)	4.9 (4.2–5.5)	0.796
Anterior–posterior (AP) (cm)	4.3 (3.7–5.2)	4.1 (3.4–4.9)	0.157
PSAD (ng/mL/cm^3^)	0.9 (0.3–2.9)	0.9 (0.3–2.8)	0.861
Gleason Score (GS)	9.0 (8–9)	9.0 (8–9)	0.092
Location			0.193
Central zone	1.6% (2/128)	3.6% (2/55)	
Transition zone	10.9% (14/128)	3.6% (2/55)	
Peripheral zone	21.1% (27/128)	14.5% (8/55)	
Multiple zone	66.4% (85/128)	78.2% (43/55)	
Rad-score	−0.542 ± 2.518	−1.503 ± 3.046	0.052

PSA: prostate-specific antigen. Prostate volume: foot–head (FH) length × right–left (RL) length × anterior–posterior (AP) length × π/6. PSAD: prostate-specific antigen density, PSA value divided by MRI-estimated prostate volume. Grading groups (GG): GG1: Gleason scores ≤ 6; GG2: Gleason scores 3 + 4; GG3: Gleason scores 4 + 3; GG4: Gleason scores 4 + 4, 3 + 5, 5 + 3; GG5: Gleason scores 4 + 5, 5 + 4, 5 + 5. *p* < 0.05 indicates a statistically significant difference.

**Table 3 cancers-15-00086-t003:** The selected radiomics features of T2WI, DWI, and T2WI + DWI models.

	Radiomics Features	Coefficient	Odds Ratio (95% CI)	*p*-Value
T2WI	T2_wavelet.HHH_glrlm_RunPercentage	−0.220	0.802 (0.533–1.236)	0.298
	T2_wavelet.HHH_ngtdm_Coarseness	1.471	4.355 (0.800–29.392)	0.106
	T2_wavelet.HLH_gldm_ SmallDependenceHighGrayLevelEmphasis	−5.081	0.006 (5.54 × 10^−6^–0.687)	0.080
	T2_wavelet.HLH_glrlm_RunPercentage	1.443	4.235 (1.481–26.510)	0.045
	T2_wavelet.HLL_ngtdm_Coarseness	−1.294	0.274 (0.043–1.324)	0.134
	T2_wavelet.LHH_gldm_ DependenceNonUniformityNormalized	5.107	1.652 (1.358–4.033)	0.104
	T2_wavelet.LHH_glszm_ SizeZoneNonUniformityNormalized	0.860	2.362 (1.187–5.205)	0.022
	T2_wavelet.LHH_ngtdm_Contrast	0.722	2.058 (1.291–3.564)	0.005
	T2_wavelet.LHL_firstorder_RootMeanSquared	0.270	1.310 (0.808–2.146)	0.268
	T2_wavelet.LLL_gldm_ SmallDependenceLowGrayLevelEmphasis	0.025	1.025 (0.637–1.626)	0.916
DWI	DWI_original_glszm_SizeZoneNonUniformityNormalized	0.378	1.460 (1.0109–2.229)	0.061
	DWI_original_shape_SurfaceArea	−0.443	0.642 (0.257–1.511)	0.324
	DWI_wavelet.HLH_glcm_MaximumProbability	−0.731	0.481 (0.272–0.763)	0.005
	DWI_wavelet.LLL_glrlm_RunLengthNonUniformity	−0.700	0.496 (0.200–1.136)	0.109
T2WI + DWI	T2_wavelet.HLH_gldm_ SmallDependenceHighGrayLevelEmphasis	0.947	2.579 (1.255–7.864)	0.030
	T2_wavelet.HLH_glrlm_RunPercentage	−0.509	0.601 (0.278–1.236)	0.176
	T2_wavelet.HLL_ngtdm_Coarseness	0.703	2.020 (0.844–6.290)	0.181
	T2_wavelet.LHH_gldm_ DependenceNonUniformityNormalized	0.834	2.303 (1.171–5.080)	0.023
	T2_wavelet.LHH_glszm_ SizeZoneNonUniformityNormalized	0.537	1.710 (1.059–2.955)	0.039
	T2_wavelet.LHH_ngtdm_Contrast	0.304	1.355 (0.808–2.315)	0.249
	T2_wavelet.LHL_firstorder_RootMeanSquared	0.343	1.409 (0.859–2.375)	0.180
	DWI_original_glszm_SizeZoneNonUniformityNormalized	0.271	1.311 (0.829–2.266)	0.289
	DWI_original_shape_SurfaceArea	−0.896	0.408 (0.162–0.896)	0.039
	DWI_wavelet.HHH_glcm_DifferenceEntropy	0.687	1.988 (1.010–4.306)	0.064
	DWI_wavelet.HLH_glcm_MaximumProbability	−0.494	0.610 (0.299–1.178)	0.151
	DWI_wavelet.HLL_gldm_ LargeDependenceLowGrayLevelEmphasis	0.377	1.457 (0.873–2.460)	0.152
	DWI_wavelet.LHH_glszm_ZoneEntropy	−0.127	0.881 (0.463–1.668)	0.697

**Table 4 cancers-15-00086-t004:** The diagnostic performance of models.

Model	Train			Test				
	AUC	Sensitivity	Specificity	P	AUC	Sensitivity	Specificity	P
Clinical	0.766 (0.698–0.834)	0.890	0.522		0.823 (0.712–0.933)	1	0.514	
T2WI	0.813 (0.753–0.873)	0.868	0.609	0.276	0.827 (0.707–0.947)	0.611	0.919	0.959
DWI	0.749 (0.678–0.819)	0.802	0.598	0.709	0.734 (0.593–0.975)	0.556	0.838	0.269
T2WI + DWI	0.879 (0.832–0.926)	0.736	0.870	0.003	0.908 (0.821–0.996)	0.944	0.811	0.197
Combined	0.906 (0.866–0.947)	0.780	0.870	<0.01	0.947 (0.884–1)	0.944	0.865	0.01

P: AUC value of T2WI model, DWI model, T2WI + DWI model, and radiomic combined clinical model, respectively, compared to AUC value of clinical model.

## Data Availability

Not applicable.

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
