# Peer review of "Predicting Tumor Perineural Invasion Status in High-Grade Prostate Cancer Based on a Clinical–Radiomics Model Incorporating T2-Weighted and Diffusion-Weighted Magnetic Resonance Images"

_cancers, 2022, doi:10.3390/cancers15010086_

Round 1

Reviewer 1 Report

Dear authors, 

the work is generally well written, but i have some questions you should better explain:

pag 1 line 22: PNI in present in 17-75% prostate cancer patients ? (citations, better explanation in introduction)

paragraph 2.2 MR image data: Is the PIRADS 2.1 protocol used? It'is the gold standard to start studies about MpMRI of prostate (better explanation of the core protocol T2 and DWI)

Pag.4 Line 13: Is the used software IBSI compliant?

Pag.4 Line 13: Could the authors report more parameters regarding feature extraction?

Author Response

Dear reviewer:We would like to take this opportunity to thank the reviewer your careful reading of our manuscript and insightful comments, which have helped improve our paper. The point-to-point responses of 4points are presented in the word document. Thanks again and looking forward to your reply as soon as possible.

Reviewer 2 Report

Summary

In this work the authors use radiomic features extracted from T2-w and DWI MR images, combined with clinical data to create a nomogram predicting perineural invasion in high-grade prostate cancer patients. This study is highly original and a great deal of attention was paid in the study design. However, some issues need to be solved before this manuscript can be considered for publication.

Comments

1)      Abstract: “The differential diagnostic efficiency of clinical model was poorer than the radiomics model (area under the curve (AUC)were 0.766 and 0.823 in train and test group respectively).” However statistical tests show no significant difference between clinical and radiomics model in the test set. Only the combined clinical + radiomics model performed better than all other investigated models.

2)      Introduction: “Prostate cancer (PCa) is the most frequent malignant tumor and the second leading cause of cancer-related death among males [1].” This only refers to the US. Please specify in the US or add a reference to worldwide data

3)      Methods section; Extraction of radiomic features: All information regarding spatial resampling, discretization and normalization of voxels prior to features’ extraction are missing. Please, add this information as it can strongly affect the results

4)      Methods section; Extraction of radiomic features: FAE has not been defined anywhere before. Is this software IBSI compliant? Are features extracted from this software somehow standardized?

5)      Methods section; Feature selection and model building: Feature selection was performed on the training set or on the whole dataset? If feature selection is performed in the whole dataset data leakage might occur therefore resulting in an overestimation of model performance (that may also explain the better results obtained in the test set as compared to training)

6)      Discussion: “In 1999, the College of American Pathologists published a consensus statement on prognostic factors for PCa in which PNI was classified as category III for the risk of recurrence and needed additional study”. Are there more recent information and literature? Please, add accordingly.

Author Response

Dear reviewer:We would like to take this opportunity to thank the reviewer your careful reading of our manuscript and insightful comments, which have helped improve our paper. The point-to-point responses of 6points are presented in the word document. Thanks again and looking forward to your reply as soon as possible.

Reviewer 3 Report

Comment to the authors

The authors developed radiomics models to predict perineural invasion (PNI) on prostate MR images of the patients with high grade prostate cancers (PCa). The model was trained with 128 patients’ data and tested with 55 patients’ data. The radiomics model showed the area under the receiver operating characteristic curve (AUC) of 0.908 in test dataset. When the model was combined with clinical model, the AUC further improved to 0.947 in test dataset. The authors concluded the combination model including radiomics and clinical models accurately predict PNI in patients with high-grade PCa. I have some queries and comments as follows.

Methods

·         Could the authors clarify why they only evaluated patients with high-grade PCa? In addition, inclusion of high-grade PCa only should be added in inclusion or exclusion criteria.

·         Line 93. Please clarify what the authors mean with “prostate MRI detection”. How did the authors define lesion visibility on prostate MRI? Were the MR images interpreted in accordance with PIRADS v 2.1?  Was the visible lesion confirmed with PCa by targeted biopsy?

·         For image interpretation, could the authors please clarify how slice levels between T2WI and DWI were matched? In addition, in the case that MRI-invisible lesion in at least one MRI sequence, how did the readers (radiologists?) delineate the resion of interest (ROI)?

Results

·         Can the authors clarify how many cases were excluded through patient selection?

·         Table 1. While some p-values less than 0.05 show up to 3 decimal point, some p-values show only “<0.05” instead of clarifying precise p-values. Could the authors show the p-values with same decimal point?

·         Table 2. Could the authors show the difference of Rad-score between Training dataset and Test dataset as shown in table 1?

·         Figure 4 need to be re-write, because some parameters are hard to be interpreted. For example, the scale of volume parameter is unclear since only “0” is labeled on the scale. GS parameter shows small increments defining 8.5 or 9.5, but GS scoring system does not define the numbers below decimal point. Since the label of Risk overlapped each other, it is hard to know the probability of PNI with the nomogram.

·         Could the authors make a table which shows the list of selected radiomics parameters with their significance? It should be useful for readers to know which radiomic parameters are most important for PNI detection.

·         The detections of PNI on biopsy or radical prostatectomy (RP) is potentially not the same. I suggest that the authors clarify how many cases were confirmed with PNI on biopsy or RP. In addition, please consider running sensitivity analysis to evaluate the impact of PNI detection method (biopsy or RP).

Discussion

·         Although the authors emphasized PNI is associated with poor prognosis, clinical significance of PNI is still controversial. Some studies including large series reported PNI is not an independent predictor of PSA recurrence (Adv Radiat Oncol. 2018 Sep 19;4(1):96-102.). Can the authors comment on the controversial clinical significance of PNI?

·         Limitation of this study should include single institutional retrospective study design without external validation.

Author Response

Dear reviewer:We would like to take this opportunity to thank the reviewer your careful reading of our manuscript and insightful comments, which have helped improve our paper. The point-to-point responses of 11points are presented in the word document. Thanks again and looking forward to your reply as soon as possible.

Round 2

Reviewer 1 Report

Dear authors,

now the work is good.